# The Adsorption of CTC onto CFBs: A Study on Fabrication of Magnetic Cellulose/Fe_3_O_4_ Beads (CFBs) and Adsorption Kinetics

**DOI:** 10.3390/ma16031189

**Published:** 2023-01-30

**Authors:** Jing Wang, Ke Shan, Yanhua Tang, Na Wu, Nan Li

**Affiliations:** School of Chemistry and Resources Engineering, Honghe University, Mengzi 661199, China

**Keywords:** magnetic cellulose/Fe_3_O_4_, adsorption, CTC, adsorption kinetics

## Abstract

Magnetic cellulose/Fe_3_O_4_ beads (CFBs) were fabricated by dispersing Fe_3_O_4_ particles in a microcrystalline cellulose (MCC) matrix. The CFBs were characterized by X-ray diffraction (XRD), vibrating sample magnetometry (VSM), energy dispersive X-ray spectrometry (EDS), Brunauer–Emmett–Teller (BET) analysis and scanning electron microscopy (SEM). The adsorption behaviors of CFBs were studied by chlortetracycline hydrochloride (CTC) adsorption experiments. By means of adsorption kinetics and isotherms, the adsorption mechanisms were explored. The results show that quasi-spherical CFBs with a BET surface area as high as 119.63 m^2^/g were successfully tailored, with the high saturation magnetization (Ms > 40 emu/g) guaranteeing the magnetic separation of CFBs from wastewater. The process of adsorbing CTC onto CFBs involves monolayer chemical adsorption, and the maximum adsorption capacity for CTC estimated by the Langmuir model is 89.53 mg/g. The CFB product shows better adsorption performance in acidic solution than in basic solution.

## 1. Introduction

Chlortetracycline hydrochloride (CTC), a broad-spectrum antibiotic, is a tetracycline antibiotic. Tetracycline antibiotics are a class of broad-spectrum antibiotics produced by actinomycetes. Over recent years, tetracycline antibiotics have been widely used as human and veterinary drugs and feed additives due to their low cost, which has led to the accumulation of tetracycline pollutants in the environment [1,2,3]. Due to the low efficiency of the biological degradation of tetracycline pollutants and the limitations of traditional water-treatment technologies, a large number of tetracycline antibiotics accumulate in the surface water, groundwater, and soil, posing potential hazards to the ecosystem and human health [4]. Therefore, it is extremely necessary and urgent to explore efficient and environmentally friendly methods for the removal of tetracycline pollutants.

Presently, to remove antibiotic residues, several methods, including coagulation, degradation, and adsorption, have already been employed [5,6]. Among these methods, the adsorption method is favored because it is highly efficient and produces no by-products [7].

Currently, the most commonly used adsorption materials include graphene, alumina, single-walled and multi-walled carbon nanotubes, zeolite, activated carbon, etc. However, the separation of these materials from water is a major challenge when they are used as adsorbents during the process of removing pollutants from contaminated water [8]. In recent years, the application of functional magnetic nanocomposites as adsorbents for the removal of pollutants from wastewater has attracted considerable attention because they possess excellent adsorption capacity and allow efficient magnetic separation [7,8,9,10].

Magnetic Cellulose/Fe_3_O_4_ Beads (CFBs) are emerging adsorbents, which are fabricated by combining porous MCC and magnetic Fe_3_O_4_ nanoparticles. CFBs have a high surface area, high stability, and good magnetic responsiveness. As a result, CFBs typically possess high adsorption capacities and can be easily separated from an aqueous solution by using an external magnetic field. The adsorption process does not generate secondary waste, and the materials involved can be recycled and facilely used on an industrial scale. The merits make CFBs a promising adsorbent for the treatment of contaminated water [11,12,13].

Magnetic cellulose/Fe_3_O_4_ beads (CFBs) use cellulose, which is biodegradable, biocompatible, and magnetically responsive, as the matrix. The study of the preparation, development, and application of new MCC materials is significantly important [14]. Liu et al. tailored a magnetic sugarcane bagasse (MSB) composite by combining citric acid-modified sugarcane bagasse (SB) and microcrystalline Fe_3_O_4_, and the adsorption capacities of this composite for Cd and Pb were 33 and 117 mg/g, respectively [15]. Sirajudheen developed a magnetic Fe_3_O_4_-enhanced GO-carboxymethyl cellulose recyclable composite to adsorb toxic azo dyes, and this composite achieved maximum adsorption capacities of 45.09, 47.32, and 46.21 mg/g for Congo red, acid red, and reactive red, respectively [16].

Shuai Peng et al. reported the preparation of Magnetic Fe_3_O_4_/cellulose microspheres with an average diameter of 100 μm by sol-gel transition method using ionic liquids (AmimCl) as the solvent for cellulose dissolution. In this study, CFBs were fabricated following the method introduced by Shuai Peng et al. by dispersing microcrystalline Fe_3_O_4_ into a matrix of cellulose dissolved in an ionic liquid. The self-prepared CFBs material is used as an adsorbent for the removal of CTC from water. The adsorption mechanisms are studied through adsorption kinetics and adsorption isotherms.

## 2. Materials and Methods

### 2.1. Materials

In this experiment, a number of reagents were employed, such as FeCl_3_·6H_2_O (Fengchuan, Tianjin, China), FeCl_2_·4H_2_O (Fengchuan, Tianjin, China), ammonia (Fengchuan, Tianjin, China), 1-methylimidazole (Aladdin, Shanghai, China), chloro-*n*-butane (Aladdin, Shanghai, China), ether, MCC (Aladdin, Shanghai, China), absolute ethanol (Hushi, Shanghai, China), Tween 80 (Fengchuan, Tianjin, China), and chlortetracycline hydrochloride (Yuanye, Shanghai, China), 1-butyl-3-methylimidazolium chloride salt ([BMIM]Cl, Aladdin, Shanghai, China), which were analytically pure.

### 2.2. Preparation of [BMIM]Cl Ionic Liquid

Chloro-*n*-butane and 1-methylimidazole at a molar ratio of 1:1.2 were mixed and heated at 70 °C in an oil bath for 48 h. The mixed solution was washed with diethyl ether (the volume of diethyl ether was one-fifth that of the mixed solution) 3 times, and the extra diethyl ether was removed with a rotary evaporator at 30 °C. The resulting ionic liquid was dried at 70 °C in a drying tank for further use.

### 2.3. The Preparation of Fe_3_O_4_ Particles

FeCl_3_·6H_2_O solution (10 mL, 0.4 mol/L) and FeCl_2_·4H_2_O solution (11 mL, 0.2 mol/L) were introduced into a three-necked flask and mixed for 10 min. Then, 50 mL liquid ammonia (1 mol/L) was added and stirred magnetically for 30 min. All stirring and reactions were conducted under a N_2_ atmosphere. The resulting black products were rinsed three times with deionized water. After magnetic separation by using an external magnetic field (and drying in a vacuum drying oven, 3.2 g Fe_3_O_4_ particles were obtained.

### 2.4. The Preparation of CFBs

We dispersed 9.6 g MCC into 96 mL [BMIM]Cl ionic liquid, and dissolution was conducted at 100 °C for 30 min in an oil bath, where a mass ratio of m(MCC):m([BMIM]Cl) = 1:10 was used. Then, Fe_3_O_4_ particles, whose mass was one-third that of MCC, were mixed with the cellulose solution and vigorously stirred for 20 min. Subsequently, the mixture was introduced into a solution consisting of vacuum pump oil and Tween 80 with a mass ratio of 2:3, which was further stirred for 2 h. The CFBs were separated from the [BMIM]Cl ionic liquid by an external magnetic field, followed by washing with absolute ethyl alcohol and deionized water 3 times each. The obtained CFBs were preserved in deionized water at 4 °C.

### 2.5. Characterization

An X-ray diffractometer (XRD, 7000LX, Shimadzu, Japan) was employed to determine the phase compositions of the samples. The microstructure and elemental distribution were observed by scanning electron microscopy (SEM, HITACHI-SU8010, Tokyo, Japan) equipped with energy dispersive spectrometry (EDS, QUANTA 400F, Hillsboro, OR, USA). The specific surface area was determined using a BET apparatus (BET, JW-BK122W, Beijing, China). A magnetometer (VSM, MPMS-XL SQUID, Les Ulis, France) was employed to measure the magnetic properties of the samples.

### 2.6. Adsorption Experiments

The adsorption properties of the CFBs were measured by CTC adsorption experiments with CTC concentrations varying from 10 to 50 mg/L. For a typical adsorption experiment, 0.1 g CFBs was added to 200 mL CTC solution without adjusting the pH value. The solution was magnetically stirred at 150 rpm and kept at 25 °C. The adsorption lasted for a total duration of 12 h, and the CTC solutions were sampled at intervals of 10 min in the first hour and 30 min in the second hour and then sampled at the third, sixth, and twelfth hour. CFBs in the samples were magnetically separated, and the CTC concentrations of the samples were analyzed by UV-vis spectrophotometry (λ = 275 nm).

The adsorption properties of CFBs at different pH values (6–12) were also investigated. The 0.1 M HCl and NaOH aqueous solutions were taken to regulate the pH of the CTC solutions.

The removal rate of CTC from the aqueous solution, the adsorption capacities of CFBs at equilibrium and at time *t* can be calculated based on Equations (1)–(3).
(1)$$D=\frac{A_{0}-A_{t}}{A_{0}}$$
where *D* is the removal rate of CTC, A_0_ represents the original absorbance, and *A_t_* means the absorbance at time *t*.
(2)$$q_{e}=\frac{\left(C_{0}-C_{e}\right)V}{m}$$
(3)$$q_{t}=\frac{\left(C_{0}-C_{t}\right)V}{m}$$
where *q_e_* and *q_t_* are the adsorption capacities for CFBs at equilibrium and at time *t*, C_0_ is the original concentration, *C_e_* is the concentration at equilibrium, and *C_t_* is the concentration at time *t*.

## 3. Results and Discussion

### 3.1. Characterization of CFBs

#### 3.1.1. XRD Results

The XRD patterns of microcrystalline Fe_3_O_4_, CFBs, and MCC are shown in Figure 1. As seen from XRD patterns of microcrystalline Fe_3_O_4_ and CFBs, the characteristic peaks at 2θ = 30.18°, 35.55°, 43.28°, 57.37°, and 62.92° corresponding to planes (220), (311), (400), (511), and (440), respectively, can be indexed to cubic-structured Fe_3_O_4_ (JCPDS Card 65-3107). The diffraction pattern of cellulose shows the typical cellulose I structure, with a sharp peak at 22.5° and a wide peak between 12° and 16°, which is basically the same as XRD patterns of cellulose reported in the references [17]. It was reported that the vanished peaks of CFBs at 20–22° indicated that cellulose had been successfully coated onto the surface of Fe_3_O_4_ [17]. The diffraction peaks at 16.8° and 22.5° in the XRD pattern of CFBs completely disappear, indicating that the cellulose was completely dissolved in the ionic liquid and successfully coated onto the surface of Fe_3_O_4_. Moreover, the intensities of all the diffraction peaks in the XRD pattern of CFBs are slightly weaker than those of the corresponding peaks of Fe_3_O_4_ particles, indicating the interaction between Fe_3_O_4_ nanoparticles and cellulose molecules [18].

#### 3.1.2. Magnetic Properties

The hysteresis loop of the CFB sample is shown in Figure 2, which shows that the saturation magnetization (M_s_) of the CFBs sample is higher than 40 emu/g. The M_s_ of CFBs in this paper is higher than other the M_s_ of magnetic bioadsorbents reported in the references, which may be attributed to the good crystallization of the as-prepared Fe_3_O_4_ nano-particles [19,20]. As revealed in the inset of Figure 2, the high magnetic intensity of the as-prepared CFBs enables their magnetic separation from wastewater using an external magnetic field when they are used as the adsorbent.

#### 3.1.3. SEM and EDS Analysis

Figure 3a,b shows the micromorphology of CFBs under different magnifications. Quasi-spherical agglomerations can be observed in the SEM images. In addition, Figure 3c shows the energy-dispersive spectrum (EDS) of cross point 2 in the SEM inset, and the accelerating voltage for EDS energy spectrum test is 5–15 kv. Surface coating using Au was applied, and the peak at around 2 KeV corresponding to Au is not identified. According to the spectrum, the elements C, O, and Fe coexist in the sample, which indicates that the magnetic cellulose microspheres have been fabricated successfully. However, the intensity of C peak is weak, and the Fe peak intensity is strong in the EDS spectrum, which may show that the Fe_3_O_4_ nanoparticles are less coated by MCC [17]. The particle sizes of CFBs are obtained by measuring at least 200 particles using SEM images. The particle size distribution of CFBs is shown in Figure 3d, where the average grain size of CFBs is calculated to be 25.32 nm.

In addition, the BET test shows that the BET surface area is 119.63 m^2^/g, the pore size is 9.21 nm, and the total pore volume of the CFBs is 0.26 cm^3^/g. The high BET value guarantees the high adsorption capacity of the CFBs as an adsorbent.

### 3.2. Adsorption Properties of the CFBs

#### 3.2.1. Adsorption Kinetics

The adsorption behaviors of CFBs under different initial CTC concentrations are shown in Figure 4a. Adsorption equilibrium can be reached within 720 min, and the maximum adsorption capacities corresponding to initial CTC concentrations of 10, 20, 30, 40, and 50 mg/L are 15.32, 29.21, 35.80, 51.78, and 61.57 mg/g, respectively.

The adsorption kinetics were studied to investigate the possible adsorption mechanisms. The linearized fitting results using the pseudo-first-order model and pseudo-second-order model [18,21] are shown in Figure 4b,c.

The pseudo-first-order kinetic model was first proposed by Lagergren and is the most commonly used adsorption kinetics equation, whose common linear form follows Equation (4).
(4)$$\log(q_{e}-q_{t})=\log q_{e}-\frac{k_{1}}{2.303}t$$
where *q_e_* and *q_t_* (mg/g) are the adsorption capacities at equilibrium and at time *t*, respectively. k_1_ (min^−1^) is the first-order kinetic rate constant.

The pseudo-second-order kinetics can be expressed by Equation (5).
(5)$$\frac{t}{q_{t}}=\frac{1}{k_{2}q_{e}^{2}}+\frac{t}{q_{e}}$$
where *q_e_* and *q_t_* (mg/g) are the adsorption capacities at equilibrium and at time *t*, respectively. k_2_ (g/mg·min) is the second-order kinetic rate constant.

The fitting parameters are listed in Table 1. As revealed by the values in Table 1, the correlation coefficient R^2^, which represents the correlation between the experimental and calculated values, is higher for the pseudo-second-order model than for the pseudo-first-order model. In addition, *q_e,cal_* calculated by the pseudo-second-order model agrees much better with the experimental value *q_e,exp_* than *q_e,cal_* calculated by the pseudo-first-order model. Therefore, the pseudo-second-order kinetic model describes the adsorption of CTC onto CFBs better than the pseudo-first-order model. Typically, the pseudo-second-order kinetic mode is believed to describe a chemical adsorption process [22]. The chemical property of the CTC adsorption onto CFBs is further verified by the FTIR test. The FTIR spectra of CFBs before and after CTC adsorption are shown in Figure 5. Three bands at 578, 1235, and 3423 cm^−1^ are observed in the FT-IR spectrum of CFBs. According to Shuai Peng et al., the peak at 578 cm^−1^ is assigned to the characteristic absorbance peak of Fe_3_O_4_ [18]. The peak at 1235 cm^−1^ corresponds to the C-O-C stretching vibration [23], and the peak at 3423 cm^−1^ is due to the stretching frequency of the -OH group [24]. By comparing the two spectra, it can be easily identified that the stretching vibration peaks at 578 cm^−1^ and 1235^−1^ are obviously weakened for CFBs after CTC adsorption, indicating chemical changes in the two groups. As a result, the adsorption of CTC onto CFBs is a chemical adsorption process.

In addition, with the increasing CTC concentration, the *q_e,cal_* values increase, while the k_2_ values decrease in Table 1. The higher *q_e,cal_* values can be attributed to the higher driving force for the adsorption at higher CTC concentrations, which provides a higher concentration gradient. The decreasing k_2_ values can be ascribed to the insufficient active sites on the CFBs surface, which can be quickly occupied by CTC at higher concentrations [25,26,27,28,29].

#### 3.2.2. Adsorption Isotherms

The adsorption mechanisms of CTC onto CFBs were further investigated using the Langmuir and Freundlich isotherm models, as expressed in Equations (6) and (7) [30,31,32,33].
(6)$$\frac{C_{e}}{q_{e}}=\frac{1}{{bq}_{m}}+\frac{C_{e}}{q_{m}}$$
where *q_e_* is the adsorption capacity at equilibrium, *q_m_* (mg/g) is the maximum adsorption capacity, and b is the Langmuir adsorption constant.
(7)$${\ln q}_{e}={\mathrm{lnk}}_{F}+\frac{1}{n}\ln C_{e}$$
where *q_e_* (mg/g) is the capacity at equilibrium, k_F_ is the Freundlich adsorption constant, and n is the Freundlich exponential coefficient.

The linear fitting results using Langmuir and Freundlich isotherm models are shown in Figure 6. The corresponding isotherm parameters are listed in Table 2. It is suggested that the correlation coefficient R^2^ value is higher for the Langmuir model than for the Freundlich model. Therefore, the Langmuir model better describes CTC adsorption on CFB than the Freundlich model. It is well known that the Langmuir model is based on monolayer adsorption processes [34,35], where all sites on the adsorbent surface have the same size and shape and are energetically equivalent [36]. In addition, the maximum adsorption capacity (*q_m_*) calculated using the Langmuir isotherm model reaches 89.53 mg/g, indicating the excellent adsorption property of CFBs for CTC.

Combining the FTIR test, adsorption kinetics study, and adsorption isotherm study, we can come to the conclusion that the adsorption of CTC onto the as-prepared CFBs is a monolayer chemisorption process.

#### 3.2.3. Adsorption Behaviors at Different pH Values

The adsorption behavior of CTC onto CFBs under different pH values (6–12) was investigated, and the adsorption capacities of CFBs for CTC under different pH values are shown in Figure 7. When pH = 6, the adsorption capacity of CFBs reaches the highest value, and the adsorption capacity after adsorption for 2 h is 31.89 mg/g. With increasing pH value, the adsorption capacity decreases significantly to 21.78 (pH = 9) and 17.81 mg/g (pH = 9). It is suggested that CFBs exhibit better adsorption performance in acidic solutions than in basic solutions. When the pH value is higher than 7, the HTC^2−^ anions are dominant in the CTC solution. As a result, the competitive adsorption between HTC^2−^ and OH^−^ leads to adsorption decline of CTC [37,38,39].

## 4. Conclusions

In this work, magnetic cellulose/Fe_3_O_4_ beads (CFBs) were successfully prepared by dispersing Fe_3_O_4_ particles in an MCC matrix. The adsorption behaviors of this magnetically separable adsorbent for CTC were studied, and the adsorption mechanisms were explored by adsorption kinetics and isotherms. The results show that the adsorption of CTC onto CFBs involves monolayer chemical adsorption, and the maximum adsorption capacity for CTC estimated by the Langmuir model is 89.53 mg/g. The CFB product shows better adsorption performance in acidic solution than in basic solution.

However, the adsorption mechanism needs further investigation to clarify problems, such as the pH dependence adsorption capacity, and the differences of the surfaces before and after CTC adsorption should be monitored. In addition, it is also very challenging to regenerate the CFBs while maintaining their magnetic separation. These problems will be the focus of our future study.

## Figures and Tables

**Figure 1 materials-16-01189-f001:**
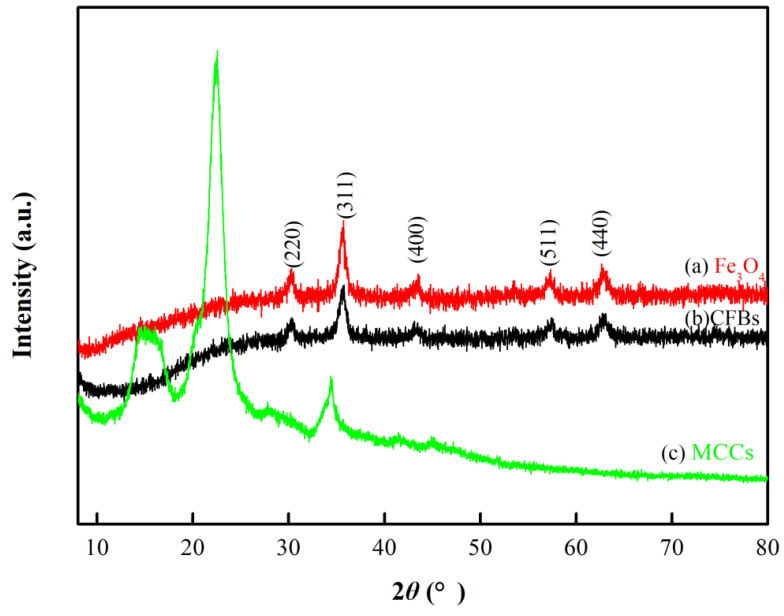
XRD patterns of (**a**) Fe_3_O_4_, (**b**) CFBs, and (**c**) MCC.

**Figure 2 materials-16-01189-f002:**
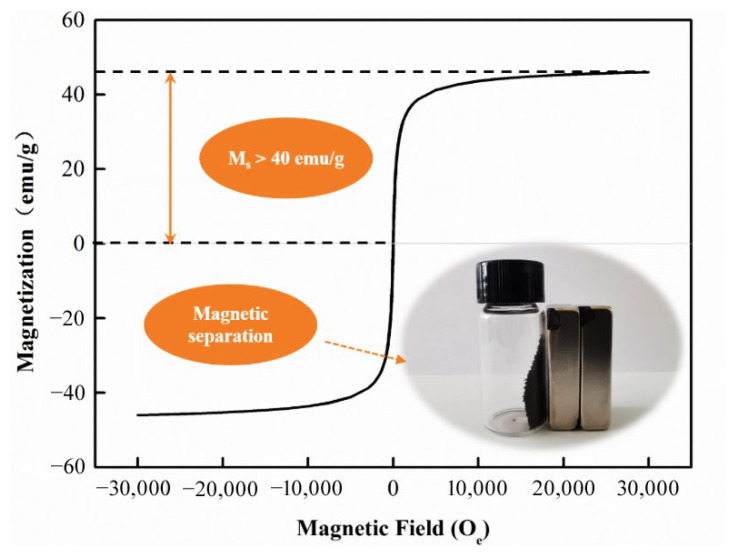
VSM analysis of CFBs.

**Figure 3 materials-16-01189-f003:**
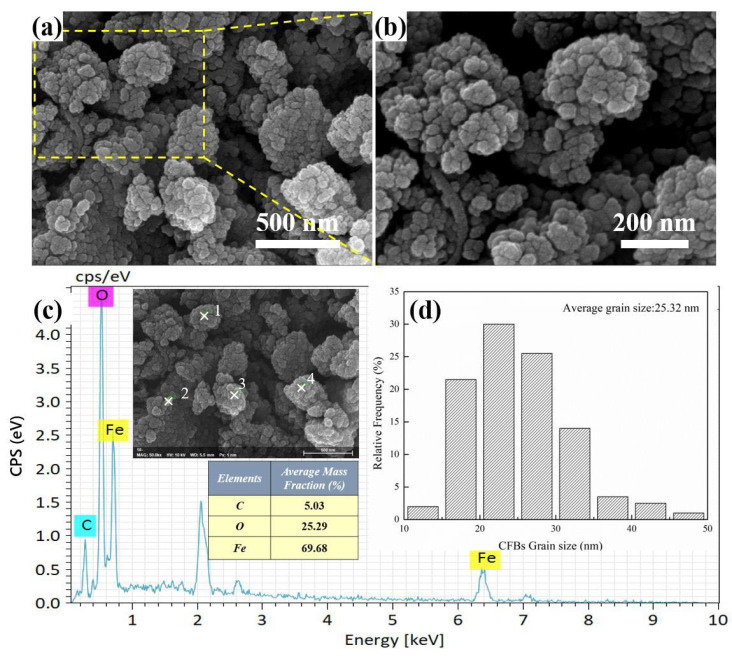
(**a**,**b**) SEM images of CFBs, (**c**) EDS spectrum of CFBs, (**d**) particle size distribution of CFBs.

**Figure 4 materials-16-01189-f004:**
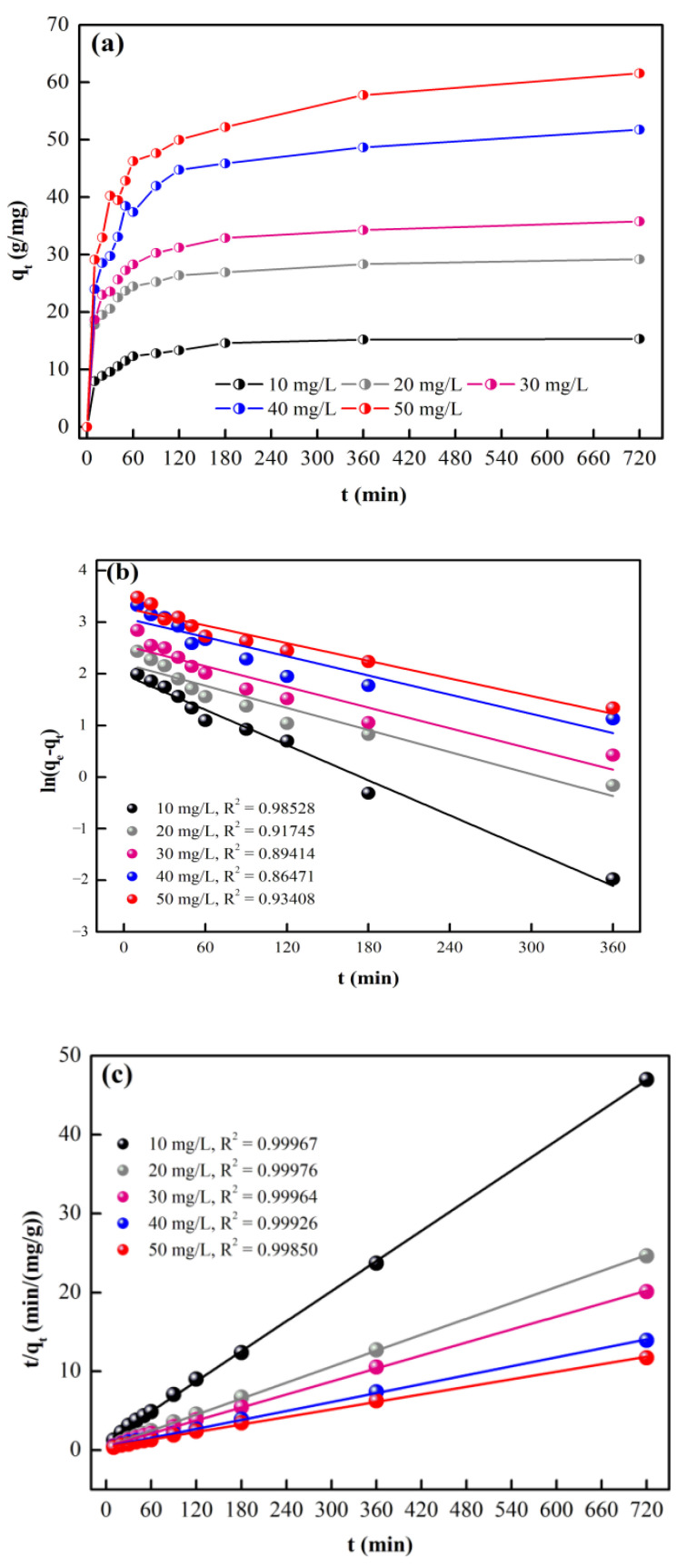
(**a**) History of adsorption capacities under different initial CTC concentrations, (**b**) linear plot of pseudo-first-order kinetics, and (**c**) linear plot of pseudo-second-order kinetics.

**Figure 5 materials-16-01189-f005:**
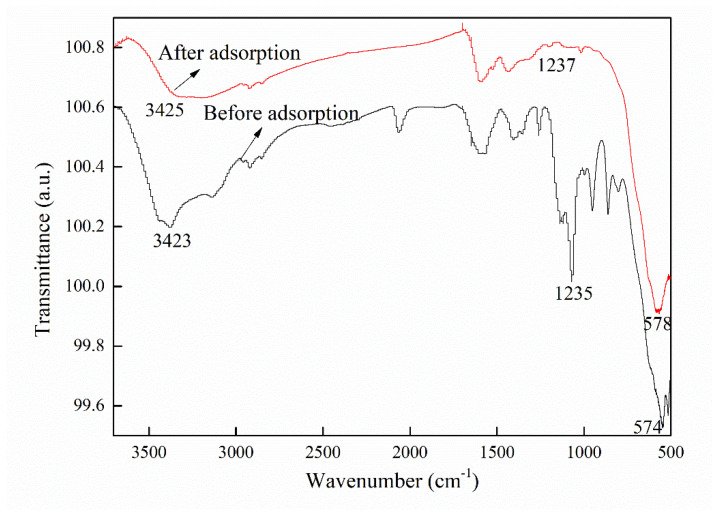
FTIR spectra of CFBs before and after CTC adsorption.

**Figure 6 materials-16-01189-f006:**
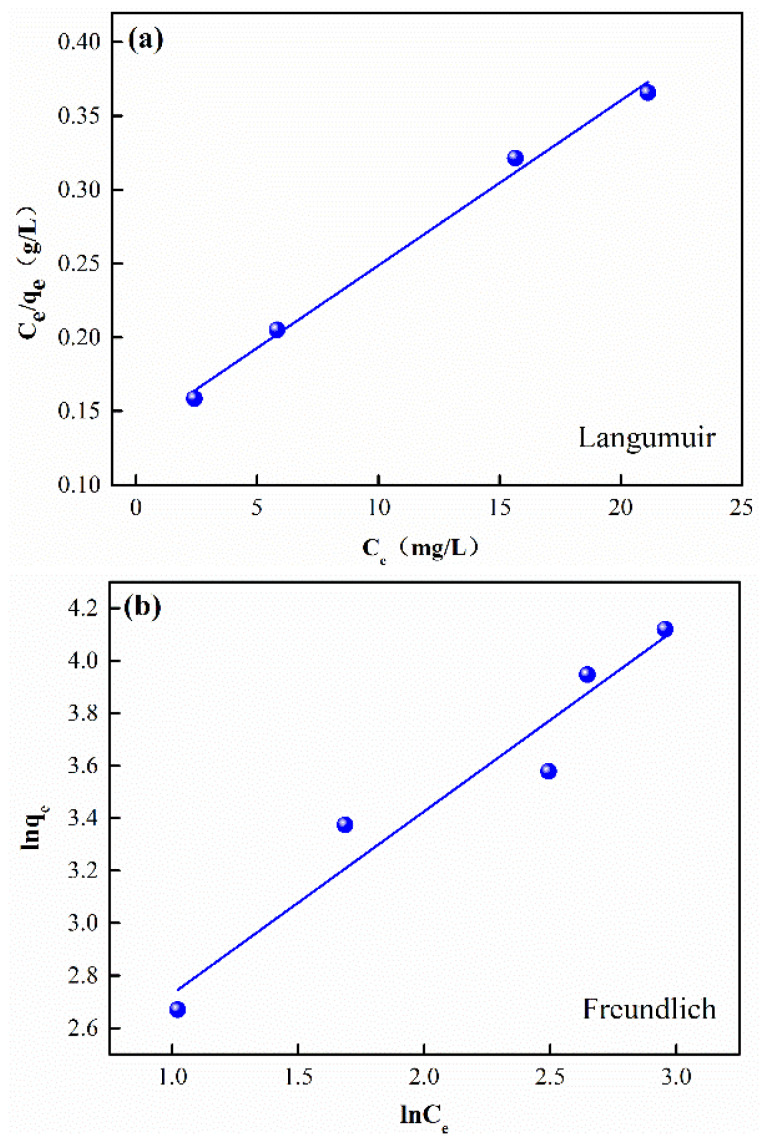
(**a**) Linear plot of the Langmuir adsorption isotherm, (**b**) linear plot of the Freundlich adsorption isotherm.

**Figure 7 materials-16-01189-f007:**
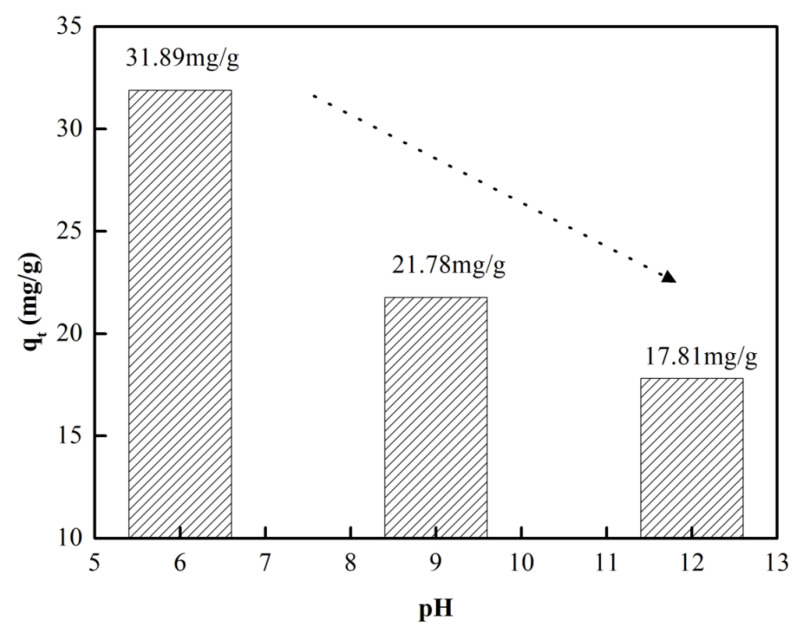
Adsorption capacities of CFBs for CTC at different pH values.

**Table 1 materials-16-01189-t001:** Fitting parameters of the first-order and second-order adsorption models.

*q_e,exp_* (mg/g)	Pseudo-First-Order Model			Pseudo-Second-Order Model		
	*q_e,cal_* (mg/g)	k_1_ (min^−1^)	R_2_	*q_e,cal_* (mg/g)	k_2_ (g/mg·min)	R_2_
15.32	7.28	1.14 × 10^−2^	0.98528	15.71	6.37 × 10^−3^	0.99967
29.21	8.99	0.71 × 10^−2^	0.91745	29.60	3.38 × 10^−3^	0.99976
35.80	12.79	0.67 × 10^−2^	0.89414	36.39	2.75 × 10^−3^	0.99964
51.78	21.81	0.62 × 10^−2^	0.86471	52.88	1.89 × 10^−3^	0.99926
61.57	26.51	0.57 × 10^−2^	0.93408	62.85	1.59 × 10^−3^	0.99850

**Table 2 materials-16-01189-t002:** Isotherm parameters for CTC adsorption onto CFBs.

Langmuir			Freundlich		
*q_m_* (mg/g)	B (L/mg)	R^2^	n	k_F_ (L/mg)	R^2^
89.53	0.08	0.99026	1.44	7.65	0.92119

## Data Availability

No new data was created or analyzed in this study. Data sharing is not applicable to this article.

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
