# Peer review of "The Adsorption of CTC onto CFBs: A Study on Fabrication of Magnetic Cellulose/Fe3O4 Beads (CFBs) and Adsorption Kinetics"

_materials, 2023, doi:10.3390/ma16031189_

Round 1
Reviewer 1 Report
1. Too many citations are found in the Introduction section. However, almost no citations are found in the results and discussion section. This is poor writing quality.
2. Lines 62 and 63. The abbreviations can be used throughout the manuscript once they have been mentioned.
3. The materials and method section is very unclear. This is because the authors did not provide a detailed explanation of the preparation of the adsorbent. The volume of chemicals, deionized water and cellulose solution, and the mass of MCC were not clearly stated. Thus, readers might find it hard to repeat the procedures.
4. The authors should provide the equations for qt, and citations for the Langmuir and Freundlich.
5. The discussion of findings looks very short or merely a reporting style. This style of writing should be avoided. Also, one could hardly find citations in the discussion section.
6. The figures should be placed below the text, not vice versa.
7. Figure 4 (a-c) should not have a title in each plot.
8. Where are citations [26] to [28]? They are not found in the reference list.
9. The authors conclude that chemical adsorption was involved in the CTC adsorption on CFBs based on the kinetic plots. This is incorrect. The mechanism of adsorption should not solely rely on the kinetic equations. As the FTIR analysis was not conducted, concluding that chemical adsorption as the adsorption mechanism is questionable.
10. There are missing citations for the Langmuir and Freundlich equations.
11. The values of rate constants (k1 and k2) are very high and suspicious.
Author Response
RESPONSE TO REVIEWERS
Manuscript ID: Materials-1942538 “The Adsorption of CTC onto CFBs: A Study on Fabrication of MagneticCellulose/Fe3O4 Beads (CFBs) and Adsorption Kinetics” by Jing Wang, Ke Shan, Yanhua Tang, Na Wu and Nan Li.
15th of November of 2022
We would like to thank the reviewers for their thoughtful review of the manuscript. They raise important issues and their inputs are very helpful for improving the manuscript. We agree with almost all their comments and we have revised our manuscript accordingly.
We are already crafting a revised version of the paper that it states the hypothesis and the implications of our work more clearly than before. Moreover, we are including all reviewers’ suggestions and clarifying the text when needed. We respond below in detail to each of the reviewer’s comments. We hope that the reviewers will find our responses to their comments satisfactory, and we are willing to finish the revised version of the manuscript including any further suggestion that the reviewers may have.
Please, find below the referees’ comments repeated in italics and our responses inserted after each comment.
Looking forward hearing from you soon.
Sincerely,
Jing Wang, Ke Shan, Yanhua Tang, Na Wu and Nan Li
_________________________________________________________________________
Reviewer 1
- Too many citations are found in the Introduction section. However, almost no citations are found in the results and discussion section. This is poor writing quality.
Thanks for the reviewer’s good suggestions. The experimental results are discussed in more details, and the corresponding references are cited in the results and discussion section. The revised contents are highlighted in the manuscript.
- Lines 62 and 63. The abbreviations can be used throughout the manuscript once they have been mentioned.
Thank you so much for your kind suggestion. The abbreviation MCC has been used throughout the manuscript except for the first time it presents in the manuscript.
“Magnetic cellulose/Fe3O4 beads (CFBs) were fabricated by dispersing Fe3O4 particles in a microcrystalline cellulose (MCC) matrix.”
“In this experiment, a great number of reagents were employed, such as FeCl3·6H2O (Tianjin fengchuan), FeCl2·4H2O (Tianjin fengchuan), ammonia (Tianjin fengchuan), 1-methylimidazole (Aladdin), chloro-n-butane (Aladdin), ether, MCC (Aladdin),”
“In this work, magnetic cellulose/Fe3O4 beads (CFBs) were successfully prepared by dispersing Fe3O4 particles in a MCC matrix.”
- The materials and method section is very unclear. This is because the authors did not provide a detailed explanation of the preparation of the adsorbent. The volume of chemicals, deionized water and cellulose solution, and the mass of MCC were not clearly stated. Thus, readers might find it hard to repeat the procedures.
Thanks for your kind reminding. The detailed dosages of the raw materials have been provided in the manuscript now.
“9.6g MCCs were dispersed into 96 mL [BMIM]Cl ionic liquid, and dissolution was conducted at 100 °C for 30 min in an oil bath, where a mass ratio of m(MCCs):m([BMIM]Cl) = 1:10 was used.”
- The authors should provide the equations for qt, and citations for the Langmuir and Freundlich.
The authors appreciate your kind suggestion. The equation for qt has been provided as equation (3), citations for the Langmuir and Freundlich models have been added in the manuscript.
“ (3)
where qe and qt are the adsorption capacities for CFBs at equilibrium and at time t, C0 is the original concentration, Ce is the concentration at equilibrium, and Ct is the concentration at time t.”
“The adsorption mechanisms of CTC onto CFBs were further investigated using the Langmuir and Freundlich isotherm models, as expressed in Equations (6) and (7) [37-39].”
- The discussion of findings looks very short or merely a reporting style. This style of writing should be avoided. Also, one could hardly find citations in the discussion section.
Thanks for the reviewer’s good suggestions. We have increased discussion and citations in the results and discussion section. The relative contents are highlighted in the manuscript.
- The figures should be placed below the text, not vice versa.
We are sorry for our mistakes. All the figures are placed below the text now.
- Figure 4 (a-c) should not have a title in each plot.
Thanks for the reviewer’s good suggestions. We have removed title in each plot of the original figure 4 (a-c).
Fig.4 (a) History of adsorption capacities under different initial CTC concentrations, (b) linear plot of pseudo-first-order kinetics, and (c) linear plot of pseudo-second-order kinetics.
- Where are citations [26] to [28]? They are not found in the reference list.
We apologize for the mistakes. Citations [26] to [28] have been added in the manuscript. The citations throughout the manuscript have been checked.
- The authors conclude that chemical adsorption was involved in the CTC adsorption on CFBs based on the kinetic plots. This is incorrect. The mechanism of adsorption should not solely rely on the kinetic equations. As the FTIR analysis was not conducted, concluding that chemical adsorption as the adsorption mechanism is questionable.
Thanks for the reviewer’s good suggestions. FTIR tests have been conducted on CFBs before and after CTC adsorption. The results are provided in Fig.6 and discussed.
The chemical property of the CTC adsorption onto CFBs is further verified by the FTIR test. The FTIR spectra of CFBs before and after CTC adsorption are shown in Fig.5. The peaks at 578 cm-1 and 3423 cm-1 are assigned to the stretching vibration F-O of Fe3O4 [33]. The peaks at 1235cm-1 corresponds the C-O-C stretching vibration [34]. By comparing the two spectra, it can be easily identified that the stretching vibration peaks at 578 cm-1 and 1235-1 are obviously weakened for CFBs after CTC adsorption, indicating chemical changes of the two groups. As a result, the adsorption of CTC onto CFBs is a chemical adsorption process.
Fig.5 FTIR spectra of CFBs before and after CTC adsorption
- There are missing citations for the Langmuir and Freundlich equations.
Thanks for your kind reminder. Citation [37-39] are added for the Langmuir and Freundlich equations.
- The values of rate constants (K1and K2) are very high and suspicious.
The authors apologize for their mistakes. The K1 and K2 values were put under close scrutiny and have been corrected.
Table 1 Fitting parameters of the first-order and second-order adsorption models.
|
qe,exp (mg/g) |
Pseudo-first-order model |
Pseudo-second-order model |
|||||
|
qe,cal (mg/g) |
k1 (min-1) |
R2 |
qe,cal (mg/g) |
k2 (g/mg·min) |
R2 |
||
|
15.32 |
7.28 |
1.14 × 10-2 |
0.98528 |
15.71 |
6.37 × 10-3 |
0.99967 |
|
|
29.21 |
8.99 |
0.71 × 10-2 |
0.91745 |
29.60 |
3.38 × 10-3 |
0.99976 |
|
|
35.80 |
12.79 |
0.67 × 10-2 |
0.89414 |
36.39 |
2.75 × 10-3 |
0.99964 |
|
|
51.78 |
21.81 |
0.62 × 10-2 |
0.86471 |
52.88 |
1.89 × 10-3 |
0.99926 |
|
|
61.57 |
26.51 |
0.57 × 10-2 |
0.93408 |
62.85 |
1.59 × 10-3 |
0.99850 |
|
Reviewer 2 Report
Work entitled “The Adsorption of CTC onto CFBs: A Study on Fabrication of Magnetic Cellulose/Fe3O4 Beads (CFBs) and Adsorption Kinetics” is interesting. It discusses the fabrication of Magnetic cellulose/Fe3O4 beads (CFBs) by dispersing Fe3O4 particles in a microcrystalline cellulose (MCC) matrix. However, the authors need to make some significant improvements in the manuscript.
Comments
3. Pattern of writing units should be uniform i.e., mg.g-1 or mg/g.
4. Mention the full form of [BMIM]Cl.
56. Align equations 1 to 6 properly.
7. Alignment of captions of all figures and naming pattern should be uniform.
8. Naming of tables 1 and 2 should either be bold or not (should be uniform).
9. In name of figure 1, “XRD patterns of (a) Fe3O4, CFBs, and MCCs.” Indicate the (a) in figure.
10. As shown in Fig. 2 Fig 1, strong diffraction peaks at 16.8° and 24.6° can be identified in the XRD pattern of MCCs.
11. “However, the diffraction peaks at 16.8° and 24.6° completely disappear in the XRD pattern”, XRD pattern of which compound? Mention clearly with name.
12. Mention the y-axis of figure 3.
13. Align fig 4 a, b and c properly. Mention the units of y-axis of figure 4b.
14. Review the units of axis of the figures properly.
15. Mention the units of adsorption capacity with mentioned values in figure 6.
16. Align all figures properly and equally.
Most importantly authors should provide all raw data files (excel files) as supplementary data, especially for the Adsorption isotherms and kinetics studies.
Author Response
RESPONSE TO REVIEWERS
Manuscript ID: Materials-1942538 “The Adsorption of CTC onto CFBs: A Study on Fabrication of MagneticCellulose/Fe3O4 Beads (CFBs) and Adsorption Kinetics” by Jing Wang, Ke Shan, Yanhua Tang, Na Wu and Nan Li.
15th of November of 2022
We would like to thank the reviewers for their thoughtful review of the manuscript. They raise important issues and their inputs are very helpful for improving the manuscript. We agree with almost all their comments and we have revised our manuscript accordingly.
We are already crafting a revised version of the paper that it states the hypothesis and the implications of our work more clearly than before. Moreover, we are including all reviewers’ suggestions and clarifying the text when needed. We respond below in detail to each of the reviewer’s comments. We hope that the reviewers will find our responses to their comments satisfactory, and we are willing to finish the revised version of the manuscript including any further suggestion that the reviewers may have.
Please, find below the referees’ comments repeated in italics and our responses inserted after each comment.
Looking forward hearing from you soon.
Sincerely,
Jing Wang, Ke Shan, Yanhua Tang, Na Wu and Nan Li
- Pattern of writing units should be uniform i.e., mg.g-1or mg/g.
Thank you for the kind reminder. All the mg.g-1 in the manuscript have been replaced by mg/g.
“and the adsorption capacities of this composite for Cd and Pb were 33 mg/g and 117 mg/g”
- Mention the full form of [BMIM]Cl.
Thanks you for your kind suggestion. The full form of [BMIM]Cl has been mentioned in Section 2.1.
“1-butyl-3-methylimidazolium chloride salt ([BMIM]Cl, Aladdin)”
- Align equations 1 to 6 properly.
Thanks for the reviewer’s good suggestions. We have Align equations 1 to 7. a
- Alignment of captions of all figures and naming pattern should be uniform.
Thank you for kindly reminding. All the figure titles have been named uniformly and aligned to the center.
- 5. Naming of tables 1 and 2 should either be bold or not (should be uniform).
We are sorry for the mistakes. The bold form has been canceled for captions of tables 1 and 2.
- In name of figure 1, “XRD patterns of (a) Fe3O4, CFBs, and MCCs.” Indicate the (a) in figure.
Thank you for your kind reminder. We have indicated the (a), (b), (c) in figure 1.
Fig.1 XRD patterns of (a) Fe3O4, (b)CFBs, and(c) MCC.
- As shown in Fig. 1, strong diffraction peaks at 16.8° and 24.6° can be identified in the XRD pattern of MCCs. “However, the diffraction peaks at 16.8° and 24.6° completely disappear in the XRD pattern”, XRD pattern of which compoundMention clearly with name.
Thanks for kindly reminding. The XRD pattern belongs to CFBs. We have mentioned it in the manuscript following your suggestion.
“However, the diffraction peaks at 16.8° and 24.6° completely disappear in the XRD pattern of CFBs”
- Mention the y-axis of figure 3.
Thanks for the reviewer’s good suggestions. We have the y-axis title has been added in figure 3 (c).
Fig.3 (a) and (b) SEM images of CFBs, (c) EDS spectrum of CFBs, (d) particle size distribution of CFBs
- Align Fig 4 a, b and c properly. Mention the units of y-axis of figure 4b.
Thank you for your kind suggestion. Fig. 4 a, b and c have been aligned to the center. The authors have discussed the units of y-axis of Fig.4 b carefully, and a number of corresponding references have been referred to. It seems that no reference uses unit for ln(qe-qt).
- Review the units of axis of the figures properly.
Thank you for kindly reminding. Units of axis of all the figures have been checked carefully to ensure that all the figures are correctly expressed.
- Mention the units of adsorption capacity with mentioned values in figure 6.
Thank you for your kind reminder. The units of adsorption capacities have been added in Fig. 6.
Fig.8 Adsorption capacities of CFBs for CTC at different pH values.
- 12. Align all figures properly and equally.
Thanks you for your kind suggestion. We have tried our best to align all the figures properly.
- 13.Most importantly authors should provide all raw data files (excel files) as supplementary data, especially for the Adsorption isotherms and kinetics studies.
Thanks for the reviewer’s good suggestions. We have provided all raw data files (excel files) as supplementary data.

Reviewer 3 Report
These, in my opinion, are the points to reconsider before classifying the work and recommending its publication:
●The characterization of magnetic microspheres fabricated by dispersion of Fe3O4 into a cellulose matrix dissolved in an ionic liquid was reported in a ‘very similar’ manner in this online article (which also cites other literature reports on the subject): ‘Advanced Materials Research 'Characterization of Magnetic Cellulose Microspheres Reconstituted from Ionic Liquid Authors: Shuai Peng, Juan Fan, Jie Chang Vols. 634-638 pp 913-917 (2013) Online: 2013-01-11’
The authors should refer to it and compare the characteristic of the obtained CFBs reported in their work.
● Regarding the adsorption of CTC onto CFBs the authors properly refer to literature and report their study on the adsorption behaviour which proves to be based on monolayer chemical adsorption. Given the importance of practical applications with real samples, the authors should give some anticipations, at least as a perspective for future work, on the adsorption mechanism which justifies the pH dependence, on the use of proper techniques to monitor the surfaces before and after CTC adsorption, the influence of eventual interferents and the possibility to regenerate the microspheres while maintaining their magnetic separation…. and so on, that is a necessary comparison with the other methods devoted to removal of CTC from wastewater.
● some corrections required
-line 129 …As shown in Fig. 1
-all figure captions should be more detailed
- the EDS spectrum to which of the four points marked with a cross does it refers and to which accelerating voltage? Why is the peak at around 2KeV (most likely due to the surface coating) not labeled?
-the list of references ends at number 25- but then [28] is cited on line 167 and [29] on line 202- skipping 26 and 27 in the numbering and both not added in the list.
Author Response
RESPONSE TO REVIEWERS
Manuscript ID: Materials-1942538 “The Adsorption of CTC onto CFBs: A Study on Fabrication of MagneticCellulose/Fe3O4 Beads (CFBs) and Adsorption Kinetics” by Jing Wang, Ke Shan, Yanhua Tang, Na Wu and Nan Li.
15th of November of 2022
We would like to thank the reviewers for their thoughtful review of the manuscript. They raise important issues and their inputs are very helpful for improving the manuscript. We agree with almost all their comments and we have revised our manuscript accordingly.
We are already crafting a revised version of the paper that it states the hypothesis and the implications of our work more clearly than before. Moreover, we are including all reviewers’ suggestions and clarifying the text when needed. We respond below in detail to each of the reviewer’s comments. We hope that the reviewers will find our responses to their comments satisfactory, and we are willing to finish the revised version of the manuscript including any further suggestion that the reviewers may have.
Please, find below the referees’ comments repeated in italics and our responses inserted after each comment.
Looking forward hearing from you soon.
Sincerely,
Jing Wang, Ke Shan, Yanhua Tang, Na Wu and Nan Li
- The characterization of magnetic microspheres fabricated by dispersion of Fe3O4into a cellulose matrix dissolved in an ionic liquid was reported in a ‘very similar’ manner in this online article (which also cites other literature reports on the subject): ‘Advanced Materials Research 'Characterization of Magnetic Cellulose Microspheres Reconstituted from Ionic Liquid Authors: Shuai Peng, Juan Fan, Jie Chang Vols. 634-638 pp 913-917 (2013) Online: 2013-01-11’The authors should refer to it and compare the characteristic of the obtained CFBs reported in their work.
Thanks for the reviewer’s good suggestions. The introduction has been expanded, and the advantages of using CFBs in the treatment of water contamination have been clarified.
“Magnetic Cellulose/Fe3O4 Beads (CFBs) are an emerging adsorbent, which are fabricated by combining porous MCC and magnetic Fe3O4 nanoparticles. CFBs have high surface area, high stability, and good magnetic responsiveness. As a result, CFBs typically possess high adsorption capacities and can be easily separated from aqueous solution by using an external magnetic field. The adsorption process does not generate secondary waste and the materials involved can be recycled and facilely used on an industrial scale. The merits make CFBs a promising adsorbent for the treatment of contaminated water [23-26].”
- Regarding the adsorption of CTC onto CFBs the authors properly refer to literature and report their study on the adsorption behaviour which proves to be based on monolayer chemical adsorption.Given the importance of practical applications with real samples, the authors should give some anticipations, at least as a perspective for future work, on the adsorption mechanism which justifies the pH dependence, on the use of proper techniques to monitor the surfaces before and after CTC adsorption, the influence of eventual interferents and the possibility to regenerate the microspheres while maintaining their magnetic separation…. and so on, that is a necessary comparison with the other methods devoted to removal of CTC from wastewater.
Thanks for the reviewer’s good suggestions. The revised contents are highlighted in the manuscript.
- some corrections required-line 129 …As shown in Fig. 1-all figure captions should be more detailed
Thanks for the reviewer’s good suggestions. The revised contents are highlighted in the manuscript.
- the EDS spectrum to which of the four points marked with a cross does it refers and to which accelerating voltage? Why is the peak at around 2KeV (most likely due to the surface coating) not labeled?
Thanks for the reviewer’s good suggestions. We have the y-axis title has been added in EDS spectrum.
5.-the list of references ends at number 25- but then [28] is cited on line 167 and [29] on line 202- skipping 26 and 27 in the numbering and both not added in the list.
We apologize for the mistakes. Citations [26] to [28] have been added in the manuscript. The citations throughout the manuscript have been checked.

Reviewer 4 Report
The authors prepared a very decent paper after doing a huge work. Nevertheless, I would like to give several objections:
Introduction should be expanded, concretely a paragraph starting with a sentence: "Magnetic cellulose/Fe3O4 beads (CFBs) use cellulose..." In the paragraph were mentioned three references, but I could not find the reason why the authors have chosen CFBs. Please, explain
In a subsection 2.1. Materials should be given exact list of used chemicals as well as their producers
In a subsection 2.3. The preparation of Fe3O4 particles should be described magnetic separation.
Why were not given data for Fe3O4 particles size? In Figure 3a can be concluded that particles size is 500 nm? Did the author consider a use of Fe3O4 nanoparticles?
I greet a using professional equation editor by the authors instead of a built-in.
Author Response
RESPONSE TO REVIEWERS
Manuscript ID: Materials-1942538 “The Adsorption of CTC onto CFBs: A Study on Fabrication of MagneticCellulose/Fe3O4 Beads (CFBs) and Adsorption Kinetics” by Jing Wang, Ke Shan, Yanhua Tang, Na Wu and Nan Li.
15th of November of 2022
We would like to thank the reviewers for their thoughtful review of the manuscript. They raise important issues and their inputs are very helpful for improving the manuscript. We agree with almost all their comments and we have revised our manuscript accordingly.
We are already crafting a revised version of the paper that it states the hypothesis and the implications of our work more clearly than before. Moreover, we are including all reviewers’ suggestions and clarifying the text when needed. We respond below in detail to each of the reviewer’s comments. We hope that the reviewers will find our responses to their comments satisfactory, and we are willing to finish the revised version of the manuscript including any further suggestion that the reviewers may have.
Please, find below the referees’ comments repeated in italics and our responses inserted after each comment.
Looking forward hearing from you soon.
Sincerely,
Jing Wang, Ke Shan, Yanhua Tang, Na Wu and Nan Li
- 1.Introduction should be expanded, concretely a paragraph starting with a sentence: "Magnetic cellulose/Fe3O4beads (CFBs) use cellulose..." In the paragraph were mentioned three references, but I could not find the reason why the authors have chosen CFBs. Please, explain.
Thanks for the reviewer’s good suggestions. The introduction has been expanded, and the advantages of using CFBs in the treatment of water contamination have been clarified.
“Magnetic Cellulose/Fe3O4 Beads (CFBs) are an emerging adsorbent, which are fabricated by combining porous MCC and magnetic Fe3O4 nanoparticles. CFBs have high surface area, high stability, and good magnetic responsiveness. As a result, CFBs typically possess high adsorption capacities and can be easily separated from aqueous solution by using an external magnetic field. The adsorption process does not generate secondary waste and the materials involved can be recycled and facilely used on an industrial scale. The merits make CFBs a promising adsorbent for the treatment of contaminated water [23-26].”
2.In a subsection 2.1. Materials should be given exact list of used chemicals as well as their producers.
Thank you for your kind reminder. The detailed manufacturers of the chemicals used in this study have been added in section 2.1.
“In this experiment, a great number of reagents were employed, such as FeCl3·6H2O (Tianjin fengchuan), FeCl2·4H2O (Tianjin fengchuan), ammonia (Tianjin fengchuan), 1-methylimidazole (Aladdin), chloro-n-butane (Aladdin), ether, MCC (Aladdin), absolute ethanol (Shanghai Hushi), Tween 80 (Tianjin fengchuan), and chlortetracycline hydrochloride (shanghai yuanye), 1-butyl-3-methylimidazolium chloride salt ([BMIM]Cl, Aladdin), which were analytically pure.”
- 3.In a subsection 2.3. The preparation of Fe3O4 particles should be described magnetic separation.
Thank you for kindly reminder. The magnetic separation of Fe3O4 particles was conducted by using an external magnetic field as shown in the inset of Fig.2. The corresponding information has been added in section 2.3.
“After magnetic separation by using and external magnetic field (inset of Fig.2).”
4.Why were not given data for Fe3O4 particles size? In Figure 3a can be concluded that particles size is 500 nm? Did the author consider a use of Fe3O4 nanoparticles?
Thank you for kindly reminder. The particle sizes of CFBs were measured and provided in Fig.3 d.
The particle sizes of CFBs are obtained by measuring at least 200 particles using SEM images. Particle size distribution of CFBs is shown in Fig.3 (d), where the average grain size of CFBs is calculated to be 25.32 nm.
Fig.3 (a) and (b) SEM images of CFBs, (c) EDS spectrum of CFBs, (d) particle size distribution of CFBs
- 5.I greet a using professional equation editor by the authors instead of a built-in.
Thank you for kindly reminder. We have used professional equation editor in the manuscript.

Round 2
Reviewer 1 Report
1. Lines 212 & 214. The symbol should be k2, not K2
2. Table 1. The unit min-1 should be min-1
3. Line 225 and Table 2. KF should be kF.
4. Table 2. KL should be changed to 'b', as presented in equation 6
Author Response
- Lines 212 & 214. The symbol should be k2, not K2
Thank you for the kind reminder. the symbol K2 in the Lines 212 & 214 of manuscript have been replaced by k2.
- Table 1. The unit min-1 should be min-1
Thank you for the kind reminder. the min-1 in the Table 1 of manuscript have been replaced by min-1.
- Line 225 and Table 2. KF should be kF.
Thank you for the kind reminder. the symbol KF in the Line 225 and Table 2 of manuscript have been replaced by kF.
- Table 2. KL should be changed to 'b', as presented in equation 6
Thank you for the kind reminder. the symbol KL in the Table 2 of manuscript have been replaced by kb.

Reviewer 2 Report
No more comments
Author Response

(The authors gave the same response as above.)

Reviewer 3 Report
The authors did not answer the questions I asked, here then resubmitted, also enclosing the paper mentioned in my previous report which in my opinion should have been cited in their Introduction i.e. in the paragraph related to the CFBs fabrication: (lines 64-67 of the revised version), due to the strong similarity in the procedure as already said in my first revision (although with some errors and disattentions in the text).
Then it was asked to compare their CFBs characterizations with those therein reported.
In particular, it was asked to better specify the SEM/EDS experiments and better clarify the EDS spectrum, hoping that by comparison some justifications could be provided on the different peak intensity, especially for the C peak and on the peak reported at 2 KeV not labeled (in both cases).
In fact, even comparing XRD, FTIR and VSM analyisis it seems that in CFBs fabricated by the authors, the Fe3O4 nanoparticles are perhaps ‘less coated’ by MCC. Thus the comparison would have been important in my opinion to provide information on the best procedure to propose also in terms of reproducibility, given the strong similarity in the setup of the two experiments, not found elsewhere.
Incidentally, one of the authors of the attached article was cited in the manuscript (now as reference 30) but only in the paragraph on adsorption kinetics and referring to what appears to be an internal report from the University
In summary, in consideration of parts of the manuscript that I see revised by another referee such as adsorption experiments, adsorption properties of CFBs and chemisorption mechanism of CTC onto CFBs, I ask the Editors to consider his/her judgment and to request explanations from the authors for the lack of responses for my part of the review, for the purpose of the possible acceptance of the work.

Author Response
RESPONSE TO REVIEWERS
Manuscript ID: Materials-1942538 “The Adsorption of CTC onto CFBs: A Study on Fabrication of MagneticCellulose/Fe3O4 Beads (CFBs) and Adsorption Kinetics” by Jing Wang, Ke Shan, Yanhua Tang, Na Wu and Nan Li.
30th of October of 2022
We would like to thank the reviewers for their thoughtful review of the manuscript. They raise important issues and their inputs are very helpful for improving the manuscript. We agree with almost all their comments and we have revised our manuscript accordingly.
We are already crafting a revised version of the paper that it states the hypothesis and the implications of our work more clearly than before. Moreover, we are including all reviewers’ suggestions and clarifying the text when needed. We respond below in detail to each of the reviewer’s comments. We hope that the reviewers will find our responses to their comments satisfactory, and we are willing to finish the revised version of the manuscript including any further suggestion that the reviewers may have.
Please, find below the referees’ comments repeated in italics and our responses inserted after each comment.
Looking forward hearing from you soon.
Sincerely,
Jing Wang, Ke Shan, Yanhua Tang, Na Wu and Nan Li
_________________________________________________________________________
The first review comments
1.The characterization of magnetic microspheres fabricated by dispersion of Fe3O4 into a cellulose matrix dissolved in an ionic liquid was reported in a ‘very similar’ manner in this online article (which also cites other literature reports on the subject): ‘Advanced Materials Research 'Characterization of Magnetic Cellulose Microspheres Reconstituted from Ionic Liquid Authors: Shuai Peng, Juan Fan, Jie Chang Vols. 634-638 pp 913-917 (2013) Online: 2013-01-11’The authors should refer to it and compare the characteristic of the obtained CFBs reported in their work.
Thanks for the reviewer’s good suggestions. We have cited the reference in the manuscript.
Shuai Peng et. al. reported the preparation of Magnetic Fe3O4/cellulose microspheres with an average diameter of 100 um by sol-gel transition method using ionic liquids (AmimCl) as the solvent for cellulose dissolution [18].In this stuy, CFBs were fabricated following the method introduced by Shuai Peng et. al. by dispersing microcrystalline Fe3O4 into a matrix of cellulose dissolved in an ionic liquid. The self-prepared CFBs material is used as an adsorbent for the removal of CTC from water. The adsorption mechanisms are studied through adsorption kinetics and adsorption isotherms.
2.Regarding the adsorption of CTC onto CFBs the authors properly refer to literature and report their study on the adsorption behaviour which proves to be based on monolayer chemical adsorption.Given the importance of practical applications with real samples, the authors should give some anticipations, at least as a perspective for future work, on the adsorption mechanism which justifies the pH dependence, on the use of proper techniques to monitor the surfaces before and after CTC adsorption, the influence of eventual interferents and the possibility to regenerate the microspheres while maintaining their magnetic separation…. and so on, that is a necessary comparison with the other methods devoted to removal of CTC from wastewater.
Thank you so much for your kind suggestion. The anticipations have been added in the conclusion part.
However, the adsorption mechanism needs further investigation to clarify problems, such as, the pH dependence adsorption capacity, and the differences of the surfaces before and after CTC adsorption should be monitored. In addition, it is also very challenging to regenerate the CFBs while maintaining their magnetic separation. These problems will be the focus of our future study.
3.some corrections required-line 129 …As shown in Fig. 1-all figure captions should be more detailed
Thanks for your kind reminding.we have checked all the figure captions to make sure they present the correct information.
4.the EDS spectrum to which of the four points marked with a cross does it refers and to which accelerating voltage? Why is the peak at around 2KeV (most likely due to the surface coating) not labeled?
Thanks for the reviewer’s good suggestions.we have the explained EDS spectrum.
In addition, Fig.3 (c) shows the energy-dispersive spectrum (EDS) of the cross point 2 in the SEM inset, and the accelerating voltage for EDS energy spectrum test is 5-15kv. Surface coating using Au was applied, and the peak at around 2KeV corresponding to Au is not identfied. According to the spectrum,the elements C, O, and Fe coexists in the sample, which indicates that the magnetic cellulose microspheres have been fabricated successfully. However, the intensity of C peak is weak, and the Fe peak intensity is strong in the EDS spectrum, which may show that the Fe3O4 nanoparticles are less coated by MCC [18].The particle sizes of CFBs are obtained by measuring at least 200 particles using SEM images. Particle size distribution of CFBs is shown in Fig.3 (d), where the average grain size of CFBs is calculated to be 25.32 nm.
- the list of references ends at number 25- but then [28] is cited on line 167 and [29] on line 202- skipping 26 and 27 in the numbering and both not added in the list.
We apologize for the mistakes. Citations [26] to [28] have been added in the manuscript.
_________________________________________________________________________The second review comments
- The authors did not answer the questions I asked, here then resubmitted, also enclosing the paper mentioned in my previous report which in my opinion should have been cited in their Introduction i.e. in the paragraph related to the CFBs fabrication: (lines 64-67 of the revised version), due to the strong similarity in the procedure as already said in my first revision (although with some errors and disattentions in the text).
We are sorry for our mistakes. The method in the reference suggested by the reviewer has been discussed, and the reference was cited.
- Then it was asked to compare their CFBs characterizations with those there in reported.
Thanks for the reviewer’s good suggestions. We have compared our CFBs characterizations with those there in reported in the manuscript.
The diffraction pattern of cellulose showes the typical cellulose I structure, with a sharp peak at 22.5°and a wide peak between 12°and 16°, which is bascially the same with XRD patterns of cellulose reported in the references [18]. It was reproted that the vanished peaks of CFBs at 20-22° indicated that cellulose had been successfully coated onto the surface of Fe3O4 [18]. The diffraction peaks at 16.8° and 22.5° in the XRD pattern of CFBs completely disappear, indicating that the cellulose was completely dissolved in the ionic liquid and successfully coated onto the surface of Fe3O4. Moreover, the intensities of all the diffraction peaks in the XRD pattern of CFBs are slightly weaker than those of the corresponding peaks of Fe3O4 particles, indicating the interaction between Fe3O4 nanoparticles and cellulose molecules[19].
The Ms of CFBs in this paper is is higher than other the Ms of magnetic bioadsorbents reported in the references, which may be attributed to the good crystallization of the as-prepared Fe3O4 nano-particles [20,21].
According to the spectrum, the elements C, O, and Fe coexists in the sample, which indicates that the magnetic cellulose microspheres have been fabricated successfully. However, the intensity of C peak is weak, and the Fe peak intensity is strong in the EDS spectrum, which may show that the Fe3O4 nanoparticles are less coated by MCC [18].
Three bands at 578 cm-1, 1235cm-1 and 3423 cm-1 are observed in the FT-IR spectrum of CFBs. According to Shuai Peng et. al., the peak at 578 cm-1 is assigned to the characteristic absorbance peak of Fe3O4 [19]. The peak at 1215 cm-1 corresponds the C-O-C stretching vibration [24], and the peak at 3439 cm-1 is due to the stretching frequency of the -OH group [25].
- In particular, it was asked to better specify the SEM/EDS experiments and better clarify the EDS spectrum, hoping that by comparison some justifications could be provided on the different peak intensity, especially for the C peak and on the peak reported at 2 KeV not labeled (in both cases).
Thanks for the reviewer’s good suggestions. We have compared our CFBs EDS spectrum with reference provided by the reviewer. We found that the C peak intensity is weak, while the Fe peak intensity is strong in our EDS spectrum by comparing, which shows the Fe3O4 nanoparticles are perhaps ‘less coated’ by MCC. Besides that we have the explained EDS spectrum in the manuscript.
In addition, Fig.3 (c) shows the energy-dispersive spectrum (EDS) of the cross point 2 in the SEM inset, and the accelerating voltage for EDS energy spectrum test is 5-15kv. Surface coating using Au was applied, and the peak at around 2KeV corresponding to Au is not identfied. According to the spectrum, the elements C, O, and Fe coexists in the sample, which indicates that the magnetic cellulose microspheres have been fabricated successfully. However, the intensity of C peak is weak, and the Fe peak intensity is strong in the EDS spectrum, which may show that the Fe3O4 nanoparticles are less coated by MCC [18]. The particle sizes of CFBs are obtained by measuring at least 200 particles using SEM images. Particle size distribution of CFBs is shown in Fig.3 (d), where the average grain size of CFBs is calculated to be 25.32 nm.
- In fact, even comparing XRD, FTIR and VSM analyisis it seems that in CFBs fabricated by the authors, the Fe3O4nanoparticles are perhaps ‘less coated’ by MCC. Thus the comparison would have been important in my opinion to provide information on the best procedure to propose also in terms of reproducibility, given the strong similarity in the setup of the two experiments, not found elsewhere.
Thanks for the reviewer’s good suggestions. We have compared our CFBs XRD, FTIR VSM and EDS analyisis with reference in the manuscript.
The diffraction pattern of cellulose showes the typical cellulose I structure, with a sharp peak at 22.5°and a wide peak between 12°and 16°, which is bascially the same with XRD patterns of cellulose reported in the references [18]. It was reproted that the vanished peaks of CFBs at 20-22° indicated that cellulose had been successfully coated onto the surface of Fe3O4 [18]. The diffraction peaks at 16.8° and 22.5° in the XRD pattern of CFBs completely disappear, indicating that the cellulose was completely dissolved in the ionic liquid and successfully coated onto the surface of Fe3O4. Moreover, the intensities of all the diffraction peaks in the XRD pattern of CFBs are slightly weaker than those of the corresponding peaks of Fe3O4 particles, indicating the interaction between Fe3O4 nanoparticles and cellulose molecules[19].
The Ms of CFBs in this paper is is higher than other the Ms of magnetic bioadsorbents reported in the references, which may be attributed to the good crystallization of the as-prepared Fe3O4 nano-particles [20,21].
According to the spectrum, the elements C, O, and Fe coexists in the sample, which indicates that the magnetic cellulose microspheres have been fabricated successfully. However, the intensity of C peak is weak, and the Fe peak intensity is strong in the EDS spectrum, which may show that the Fe3O4 nanoparticles are less coated by MCC [18].
Three bands at 578 cm-1, 1235cm-1 and 3423 cm-1 are observed in the FT-IR spectrum of CFBs. According to Shuai Peng et. al., the peak at 578 cm-1 is assigned to the characteristic absorbance peak of Fe3O4 [19]. The peak at 1235 cm-1 corresponds the C-O-C stretching vibration [24], and the peak at 3423 cm-1 is due to the stretching frequency of the -OH group [25].
- Incidentally, one of the authors of the attached article was cited in the manuscript (now as reference 30) but only in the paragraph on adsorption kinetics and referring to what appears to be an internal report from the University.
Thanks for the reviewer’s good suggestions. We found the reference suggested by you is really helpful, and we have cited it in the introduction, and analyzing part of the manuscript.
RESPONSE TO REVIEWERS
Manuscript ID: Materials-1942538 “The Adsorption of CTC onto CFBs: A Study on Fabrication of MagneticCellulose/Fe3O4 Beads (CFBs) and Adsorption Kinetics” by Jing Wang, Ke Shan, Yanhua Tang, Na Wu and Nan Li.
30th of October of 2022
We would like to thank the reviewers for their thoughtful review of the manuscript. They raise important issues and their inputs are very helpful for improving the manuscript. We agree with almost all their comments and we have revised our manuscript accordingly.
We are already crafting a revised version of the paper that it states the hypothesis and the implications of our work more clearly than before. Moreover, we are including all reviewers’ suggestions and clarifying the text when needed. We respond below in detail to each of the reviewer’s comments. We hope that the reviewers will find our responses to their comments satisfactory, and we are willing to finish the revised version of the manuscript including any further suggestion that the reviewers may have.
Please, find below the referees’ comments repeated in italics and our responses inserted after each comment.
Looking forward hearing from you soon.
Sincerely,
Jing Wang, Ke Shan, Yanhua Tang, Na Wu and Nan Li
_________________________________________________________________________
The first review comments
1.The characterization of magnetic microspheres fabricated by dispersion of Fe3O4 into a cellulose matrix dissolved in an ionic liquid was reported in a ‘very similar’ manner in this online article (which also cites other literature reports on the subject): ‘Advanced Materials Research 'Characterization of Magnetic Cellulose Microspheres Reconstituted from Ionic Liquid Authors: Shuai Peng, Juan Fan, Jie Chang Vols. 634-638 pp 913-917 (2013) Online: 2013-01-11’The authors should refer to it and compare the characteristic of the obtained CFBs reported in their work.
Thanks for the reviewer’s good suggestions. We have cited the reference in the manuscript.
Shuai Peng et. al. reported the preparation of Magnetic Fe3O4/cellulose microspheres with an average diameter of 100 um by sol-gel transition method using ionic liquids (AmimCl) as the solvent for cellulose dissolution [18].In this stuy, CFBs were fabricated following the method introduced by Shuai Peng et. al. by dispersing microcrystalline Fe3O4 into a matrix of cellulose dissolved in an ionic liquid. The self-prepared CFBs material is used as an adsorbent for the removal of CTC from water. The adsorption mechanisms are studied through adsorption kinetics and adsorption isotherms.
2.Regarding the adsorption of CTC onto CFBs the authors properly refer to literature and report their study on the adsorption behaviour which proves to be based on monolayer chemical adsorption.Given the importance of practical applications with real samples, the authors should give some anticipations, at least as a perspective for future work, on the adsorption mechanism which justifies the pH dependence, on the use of proper techniques to monitor the surfaces before and after CTC adsorption, the influence of eventual interferents and the possibility to regenerate the microspheres while maintaining their magnetic separation…. and so on, that is a necessary comparison with the other methods devoted to removal of CTC from wastewater.
Thank you so much for your kind suggestion. The anticipations have been added in the conclusion part.
However, the adsorption mechanism needs further investigation to clarify problems, such as, the pH dependence adsorption capacity, and the differences of the surfaces before and after CTC adsorption should be monitored. In addition, it is also very challenging to regenerate the CFBs while maintaining their magnetic separation. These problems will be the focus of our future study.
3.some corrections required-line 129 …As shown in Fig. 1-all figure captions should be more detailed
Thanks for your kind reminding.we have checked all the figure captions to make sure they present the correct information.
4.the EDS spectrum to which of the four points marked with a cross does it refers and to which accelerating voltage? Why is the peak at around 2KeV (most likely due to the surface coating) not labeled?
Thanks for the reviewer’s good suggestions.we have the explained EDS spectrum.
In addition, Fig.3 (c) shows the energy-dispersive spectrum (EDS) of the cross point 2 in the SEM inset, and the accelerating voltage for EDS energy spectrum test is 5-15kv. Surface coating using Au was applied, and the peak at around 2KeV corresponding to Au is not identfied. According to the spectrum,the elements C, O, and Fe coexists in the sample, which indicates that the magnetic cellulose microspheres have been fabricated successfully. However, the intensity of C peak is weak, and the Fe peak intensity is strong in the EDS spectrum, which may show that the Fe3O4 nanoparticles are less coated by MCC [18].The particle sizes of CFBs are obtained by measuring at least 200 particles using SEM images. Particle size distribution of CFBs is shown in Fig.3 (d), where the average grain size of CFBs is calculated to be 25.32 nm.
- the list of references ends at number 25- but then [28] is cited on line 167 and [29] on line 202- skipping 26 and 27 in the numbering and both not added in the list.
We apologize for the mistakes. Citations [26] to [28] have been added in the manuscript.
_________________________________________________________________________The second review comments
- The authors did not answer the questions I asked, here then resubmitted, also enclosing the paper mentioned in my previous report which in my opinion should have been cited in their Introduction i.e. in the paragraph related to the CFBs fabrication: (lines 64-67 of the revised version), due to the strong similarity in the procedure as already said in my first revision (although with some errors and disattentions in the text).
We are sorry for our mistakes. The method in the reference suggested by the reviewer has been discussed, and the reference was cited.
- Then it was asked to compare their CFBs characterizations with those there in reported.
Thanks for the reviewer’s good suggestions. We have compared our CFBs characterizations with those there in reported in the manuscript.
The diffraction pattern of cellulose showes the typical cellulose I structure, with a sharp peak at 22.5°and a wide peak between 12°and 16°, which is bascially the same with XRD patterns of cellulose reported in the references [18]. It was reproted that the vanished peaks of CFBs at 20-22° indicated that cellulose had been successfully coated onto the surface of Fe3O4 [18]. The diffraction peaks at 16.8° and 22.5° in the XRD pattern of CFBs completely disappear, indicating that the cellulose was completely dissolved in the ionic liquid and successfully coated onto the surface of Fe3O4. Moreover, the intensities of all the diffraction peaks in the XRD pattern of CFBs are slightly weaker than those of the corresponding peaks of Fe3O4 particles, indicating the interaction between Fe3O4 nanoparticles and cellulose molecules[19].
The Ms of CFBs in this paper is is higher than other the Ms of magnetic bioadsorbents reported in the references, which may be attributed to the good crystallization of the as-prepared Fe3O4 nano-particles [20,21].
According to the spectrum, the elements C, O, and Fe coexists in the sample, which indicates that the magnetic cellulose microspheres have been fabricated successfully. However, the intensity of C peak is weak, and the Fe peak intensity is strong in the EDS spectrum, which may show that the Fe3O4 nanoparticles are less coated by MCC [18].
Three bands at 578 cm-1, 1235cm-1 and 3423 cm-1 are observed in the FT-IR spectrum of CFBs. According to Shuai Peng et. al., the peak at 578 cm-1 is assigned to the characteristic absorbance peak of Fe3O4 [19]. The peak at 1215 cm-1 corresponds the C-O-C stretching vibration [24], and the peak at 3439 cm-1 is due to the stretching frequency of the -OH group [25].
- In particular, it was asked to better specify the SEM/EDS experiments and better clarify the EDS spectrum, hoping that by comparison some justifications could be provided on the different peak intensity, especially for the C peak and on the peak reported at 2 KeV not labeled (in both cases).
Thanks for the reviewer’s good suggestions. We have compared our CFBs EDS spectrum with reference provided by the reviewer. We found that the C peak intensity is weak, while the Fe peak intensity is strong in our EDS spectrum by comparing, which shows the Fe3O4 nanoparticles are perhaps ‘less coated’ by MCC. Besides that we have the explained EDS spectrum in the manuscript.
In addition, Fig.3 (c) shows the energy-dispersive spectrum (EDS) of the cross point 2 in the SEM inset, and the accelerating voltage for EDS energy spectrum test is 5-15kv. Surface coating using Au was applied, and the peak at around 2KeV corresponding to Au is not identfied. According to the spectrum, the elements C, O, and Fe coexists in the sample, which indicates that the magnetic cellulose microspheres have been fabricated successfully. However, the intensity of C peak is weak, and the Fe peak intensity is strong in the EDS spectrum, which may show that the Fe3O4 nanoparticles are less coated by MCC [18]. The particle sizes of CFBs are obtained by measuring at least 200 particles using SEM images. Particle size distribution of CFBs is shown in Fig.3 (d), where the average grain size of CFBs is calculated to be 25.32 nm.
- In fact, even comparing XRD, FTIR and VSM analyisis it seems that in CFBs fabricated by the authors, the Fe3O4nanoparticles are perhaps ‘less coated’ by MCC. Thus the comparison would have been important in my opinion to provide information on the best procedure to propose also in terms of reproducibility, given the strong similarity in the setup of the two experiments, not found elsewhere.
Thanks for the reviewer’s good suggestions. We have compared our CFBs XRD, FTIR VSM and EDS analyisis with reference in the manuscript.
The diffraction pattern of cellulose showes the typical cellulose I structure, with a sharp peak at 22.5°and a wide peak between 12°and 16°, which is bascially the same with XRD patterns of cellulose reported in the references [18]. It was reproted that the vanished peaks of CFBs at 20-22° indicated that cellulose had been successfully coated onto the surface of Fe3O4 [18]. The diffraction peaks at 16.8° and 22.5° in the XRD pattern of CFBs completely disappear, indicating that the cellulose was completely dissolved in the ionic liquid and successfully coated onto the surface of Fe3O4. Moreover, the intensities of all the diffraction peaks in the XRD pattern of CFBs are slightly weaker than those of the corresponding peaks of Fe3O4 particles, indicating the interaction between Fe3O4 nanoparticles and cellulose molecules[19].
The Ms of CFBs in this paper is is higher than other the Ms of magnetic bioadsorbents reported in the references, which may be attributed to the good crystallization of the as-prepared Fe3O4 nano-particles [20,21].
According to the spectrum, the elements C, O, and Fe coexists in the sample, which indicates that the magnetic cellulose microspheres have been fabricated successfully. However, the intensity of C peak is weak, and the Fe peak intensity is strong in the EDS spectrum, which may show that the Fe3O4 nanoparticles are less coated by MCC [18].
Three bands at 578 cm-1, 1235cm-1 and 3423 cm-1 are observed in the FT-IR spectrum of CFBs. According to Shuai Peng et. al., the peak at 578 cm-1 is assigned to the characteristic absorbance peak of Fe3O4 [19]. The peak at 1235 cm-1 corresponds the C-O-C stretching vibration [24], and the peak at 3423 cm-1 is due to the stretching frequency of the -OH group [25].
- Incidentally, one of the authors of the attached article was cited in the manuscript (now as reference 30) but only in the paragraph on adsorption kinetics and referring to what appears to be an internal report from the University.
Thanks for the reviewer’s good suggestions. We found the reference suggested by you is really helpful, and we have cited it in the introduction, and analyzing part of the manuscript.
